# Potential Role of T2-Weighted Kurtosis in Improving Response Prediction of Locally Advanced Rectal Cancer as Additional Tool Gained from Standard MRI Examination

**DOI:** 10.3390/biomedicines13123003

**Published:** 2025-12-08

**Authors:** Aleksandra Jankovic, Marko Ž. Daković, Milica Badza Atanasijevic, Milica Mitrovic-Jovanovic, Katarina Stosic, Dimitrije Sarac, Jelena Sisevic, Dusan Saponjski, Ivan Dimitrijević, Marko Miladinov, Jelenko Jelenković, Ljubica Lazic, Goran Barisic, Aleksandra Djuric-Stefanovic, Jelena Kovac

**Affiliations:** 1Department for Digestive Radiology, Center for Radiology, University Clinical Center of Serbia, 11000 Belgrade, Serbia; 2School of Medicine, University of Belgrade, 11000 Belgrade, Serbia; 3School of Physical Chemistry, University of Belgrade, 11000 Belgrade, Serbia; 4School of Electrical Engineering, University of Belgrade, 11000 Belgrade, Serbia; 5Innovation Center, School of Electrical Engineering in Belgrade, 11120 Belgrade, Serbia; 6Clinic for Digestive Surgery—First Surgical Clinic, University Clinical Center of Serbia, 11000 Belgrade, Serbia

**Keywords:** MRI, T2-weighted imaging, histogram analysis, kurtosis, rectal cancer, treatment response

## Abstract

**Background:** Reliable and accurate prediction of treatment response to neoadjuvant chemoradiotherapy (nCRT) in locally advanced rectal cancer (LARC) is usually demanding and continues to pose a challenge. Kurtosis as a histogram parameter calculated on T2-weighted MRI sequences might be an additional tool, as it represents a quantitative biomarker for response prediction. It is defined as a measure of distributions’ tails relative to the center of the distribution curve, which reflects tissue heterogeneity. The aim of the study was to evaluate the added value of T2-weighted kurtosis in predicting pathological response to nCRT in patients with LARC. **Methods:** a single-center cohort study included 71 patients with LARC who underwent both initial and post-nCRT MRI examinations followed by surgical resection in the form of the total mesorectal excision (TME). Histogram analysis was performed using software MIPAV (Medical Image Processing, Analysis, and Visualization, version 11.3.2, developed by the National Institutes of Health, Bethesda, MD, USA) on T2-weighted sequences, extracting kurtosis along with other histogram parameters. Pathological tumor regression grade (pTRG) in accordance with Mandard classification was considered the gold standard. Patients were classified as responders (pTRG 1–2) or non-responders (pTRG 3–5). **Results:** while other histogram parameters did not show statistically significant differences between groups, post-treatment values of kurtosis were significantly higher in responders compared to non-responders (4.28 ± 0.73 vs. 3.01 ± 0.17, *p* = 0.024). The F1 score as a classification metric (0.821) indicates an improvement in classification performance following therapy. **Conclusions:** T2-weighted kurtosis might be a significant tool in predicting pathological response to nCRT, representing a potentially valuable quantitative biomarker that could improve treatment response assessment.

## 1. Introduction

Colorectal cancer is among the most common forms of cancer, with approximately a third of all cases localized in the rectum [1]. It shows a male predominance with a male-to-female ratio of approximately 1.5:1, with a median age at diagnosis ranging from 60 to 65 years. Furthermore, the incidence of locally advanced rectal cancer (LARC) accounts for approximately 30–40% of all rectal cancer cases [2]. While early-stage tumors can be completely removed with surgery alone, the standard procedure for LARCs that are determined as a T3/T4 tumor or a tumor of any stage with positive lymph nodes, nowadays, includes the implementation of neoadjuvant chemoradiotherapy (nCRT) as a starting treatment followed by surgical resection (Figure 1) [2,3,4]. It is shown that this therapeutic approach lowers the risk of further cancer spreading and improves overall survival [5]. On histopathological enema, the pathological response to therapy classified by Mandard is the most commonly used scale. It ranges from tumor regression grade (TRG) 1, where it is only visible therapy—induced fibrosis, meaning complete response, to 5, meaning no response or progression [6].

Magnetic resonance imaging (MRI) is pivotal in managing LARC with its roles for primary staging, restaging, and surveillance [7]. It has extraordinary accuracy in the initial, pre-treatment examination, offering comprehensive insights for local staging. The European Society for Medical Oncology (ESMO) guidelines recommend high-resolution MRI as the imaging modality of choice for rectal cancer staging. Unfortunately, it is worth highlighting that, in contrast to the great precision of the initial above-mentioned magnetic resonance scan, the reliability of interpretation of the post-treatment MRI drops significantly, with reported poor agreement in staging between post-nCRT MRI and histopathological findings [8]. The reason for this drop is primarily attributed to the difficulty in differentiating therapy-induced fibrosis, edema, and inflammation from the residual tumor tissue [7,8,9]. This limitation has important clinical implications, particularly for patient selection for organ preservation strategies such as the ‘watch and wait’ approach, where accurate identification of complete responders is essential. To improve reliability, there is a demand for additional, new magnetic resonance tools which should be investigated and potentially implemented in everyday clinical practice [10].

Histogram analysis represents a quantitative approach to medical image interpretation that extracts statistical information from pixel intensity distributions within regions of interest [10]. It examines how frequently different signal intensity values appear across pixels or voxels within a specified region or volume of interest. Rather than relying solely on average intensity measurements, this technique analyzes the entire range of intensity values and their distribution characteristics. It generates various statistical metrics—including mean, median, standard deviation, skewness, kurtosis, and percentile values—that characterize tissue variability [11,12]. This methodology can be utilized across different MRI sequences and has demonstrated potential in cancer imaging for evaluating tumor features and monitoring therapeutic response. The key benefit of this approach is its capacity to measure tumor heterogeneity in an objective and consistent manner, potentially detecting subtle variations that might escape detection through visual interpretation or standard quantitative analysis [13,14].

Kurtosis is a statistical measure that quantifies the “peakedness” or “tailedness” of a probability distribution, providing information about the concentration of values around the distribution center [15]. In the context of medical imaging, kurtosis reflects tissue heterogeneity, with higher values indicating more concentrated pixel intensity distributions and lower values suggesting more dispersed, heterogeneous patterns. In other words, kurtosis reflects the degree of tissue heterogeneity at the microstructural level. Higher kurtosis values indicate more homogeneous tissue with pixel intensities tightly clustered around the mean, which in treated rectal cancer may correspond to uniform fibrotic tissue replacement following successful therapy. Conversely, lower kurtosis values suggest greater heterogeneity, potentially indicating admixed viable tumor, necrosis, and treatment effects. [15,16]. Recent studies have demonstrated that kurtosis and other histogram parameters derived from diffusion-weighted and dynamic contrast-enhanced imaging show promise in predicting treatment response in rectal cancer [14,15,16,17].

However, the potential value of T2-weighted gained histogram analysis, specifically kurtosis measurement, has been less extensively investigated despite its practical advantages for clinical implementation [18]. T2-weighted sequences present the cornerstone of rectal cancer MRI protocols, providing excellent tissue contrast for morphological assessment without requiring specialized diffusion imaging protocols, which are more often affected by artifacts, lowering image quality [19].

Hence, the purpose of this study was to explore the potential advantages of kurtosis derived from T2-weighted images as a parameter of histogram analysis in the forecasting response in patients with LARC following nCRT and to evaluate its added value compared to conventional assessment methods.

## 2. Materials and Methods

### 2.1. Study Design

This single-center retrospective cohort study was conducted from early 2020 to late 2022.

The study involved 71 patients with histologically confirmed rectal cancer. Inclusion criteria were locally advanced disease defined as T3+ and/or positive nodal status and/or MRF infiltration on initial MRI, completed neoadjuvant chemoradiotherapy with total radiation dose of 50.4 Gy in 28 fractions with 5-fluorouracil chemo potentiation, post-treatment MRI examination completed, and surgical resection with histopathological confirmation performed.

Exclusion criteria were minor patients, patients with contraindications for MRI, and technically inadequate one/both MRI examinations.

### 2.2. MRI Acquisition

All patients were examined using a 1.5 Tesla MR scanner (Signa HDxt; GE Healthcare, Milwaukee, WI, USA) before and after neoadjuvant chemoradiotherapy. Patient preparation included an enema for rectal lumen cleansing and a spasmolytic agent, hyoscine-butyl bromide (Buscopan) 20 mg intravenously. They were placed in a supine position, and a phased-array body coil was used for imaging. The MRI protocol consisted of axial and sagittal fast spin echo (FSE) T2-weighted imaging (TR/TE = 4500/100 ms; FOV: 280 × 280 mm; matrix: 384 × 256; slice thickness: 5 mm; spacing: 1 mm); oblique para-axial (perpendicular to the longitudinal axis of the tumor) FSE T2-weighted imaging (TR/TE = 5000/110 ms; matrix = 512 × 512; FOV = 180 × 180 mm; slice thickness = 3 mm; spacing: 0 mm); para-coronal FSE T2-weighted imaging (TR/TE = 5000/110 ms; matrix = 384 × 384; FOV = 200 × 200 mm; slice thickness = 3 mm; spacing: 0 mm); and para-axial diffusion-weighted imaging (TR/TE = 7000/70 ms; matrix = 128 × 128; FOV = 280 × 280 mm; slice thickness: 5 mm; spacing: 1 mm; b-values = 0–2000 s/mm^2^; NEX = 6).

### 2.3. MRI Technique

In accordance with widely recommended rectal cancer protocols, initial MRI is performed two weeks after histopathological verification while post-treatment MRI scan was completed 6 to 8 after the completion of chemoradiotherapy as it is shown that this time frame is needed for full effect of the radiation therapy as well as for potential side effects including edema and inflammation to improve significantly (Figure 2) [9,20].

Using software MIPAV (Medical Image Processing, Analysis, and Visualization) version 11.3.2, developed by the National Institutes of Health, Bethesda, MD, USA, histogram analysis was performed on both MRI examinations. All tumor segmentations were performed by a board-certified GI radiologist with 8 years of experience in rectal cancer imaging, blinded to pathological results. Segmentation was performed on T2-weighted axial sequences with regions of interest manually drawn on each slice containing visible tumor while carefully excluding surrounding fat, mesorectal fascia, and bowel lumen. It is followed by volume of interest (VOI) creation and automatically histogram parameter calculation (Figure 3, Figure 4 and Figure 5). The following 7 histogram parameters were measured: mean (average pixel intensity value in the region), standard deviation (measure of pixel intensity variation), median (middle pixel intensity value), kurtosis (measure of distribution “peakedness”), skewness (measure of pixel distribution asymmetry), and 5th and 95th percentiles (markers of distribution extremes). Other than the above-mentioned, other measured morphological, functional, and volumetric parameters are beyond the scope of this study.

To ensure consistency, a standardized segmentation protocol established prior to the study was followed. The radiologist had access to all available imaging sequences and clinical information necessary for accurate delineation. Quality control was performed by the same radiologist. A subset of 30 randomly selected cases was re-segmented after a washout period of 8 weeks to assess intra-observer reliability. The radiologist was blinded to their initial segmentation. Intra-observer agreement was evaluated using the intraclass correlation coefficient (ICC) and Dice similarity coefficient (DSC), demonstrating excellent reproducibility (ICC = 0.85; DSC = 0.80).

Magnetic resonance tumor regression grade (mrTRG) was assessed using the MERCURY group scoring system [21], analogous to pathological tumor regression grade (pTRG) in accordance with Mandard classification, which served as the gold standard, with TRG 1 meaning complete response, TRG 2—near complete response with dense fibrosis, TRG 3—moderate response with more than 50% of fibrosis, TRG4—poor response with subtle areas of fibrosis; TRG 5—no response or progression [6]. 

Patients were then divided into two groups, responders (pTRG 1–2) and non-responders (pTRG 3–5). This type of classification was made based on the results of recent studies showing that patients with complete and near/complete responses may benefit from an organ-preserving and „watch and wait” strategy [22,23].

### 2.4. Statistical Analysis

Descriptive and analytical statistical methods were used in this study. Among the descriptive methods, measures of central tendency (arithmetic mean) and measures of variability (standard deviation, standard error) were applied. Categorical data were presented using absolute and relative frequencies. Independent sample *t*-tests, as an exploratory method, were employed to compare parameters between responder and non-responder groups, appropriate for comparing two independent groups with continuous variables. Diagnostic accuracy involved sensitivity, specificity, positive and negative predictive values, and F1 score. Statistical analyses were performed using Python 3.10. with the usage of the following libraries, SciPy for statistical tests, scikit-learn and stastmodels.api for metrics, as well as NumPy, pandas, and matplotlib for data management and visualization.

## 3. Results

### 3.1. Patient Characteristics and Treatment Response

This study involved 71 patients diagnosed with rectal carcinoma. Of these, 66.2% (two-thirds) were male, with a mean age of 61.45 ± 11.4 years. Based on histopathological evaluation of treatment outcome, 22 patients (31%) were categorized as responders (pTRG1–2), whereas 49 patients (69%) were classified as non-responders (pTRG3–5).

### 3.2. Histogram Analysis of T2-Weighted Parameters in Relation to Treatment Response

The T2-weighted histogram analysis contained seven parameters: mean, std, skewness, kurtosis, median, 5th percentile, and 95th percentile. We observed all seven parameters in relation to treatment response. It can be noted that the mean value, median, and 95th percentile show statistically significant differences in relation to treatment response when we observe all patients (responders and non-responders) together (Figure 6).

Regarding the difference between parameters in relation to age/gender, no significant differences were found, which suggested that they could be used in the same database for further analysis (Figure 7 and Figure 8).

In terms of performed histogram analysis before and after nCRT in relation to the outcome, a statistically significant difference is observed in kurtosis values following nCRT between the examined groups, with responders demonstrating significantly higher kurtosis values compared to non-responders (*p* = 0.024) (Table 1). No statistically significant differences were found in the remaining parameters before and after nCRT in relation to the outcome (*p* > 0.05) (Table 1). Because of this exploratory statistic, we further consider only kurtosis as a potential parameter.

The Receiver Operating Characteristic (ROC) curve, with the Youden index used for selecting the optimal cutoff point, for kurtosis before and after therapy, demonstrates the results that are lower compared to random classification (Figure 9). However, such outcomes are not uncommon in the context of imbalanced datasets (responders—22, non-responders—49, ratio 1:2.2). This phenomenon can occur because the ROC metric may be less robust in scenarios where class distributions are highly skewed; it gives equal weight to both classes regardless of their prevalence; therefore, it is suboptimal for visual representation. The evaluation metrics for the classification of responders and non-responders, both before and after therapy, are shown in Table 2. Notably, the F1 score, as a metric that accounts for imbalanced data, indicates an improvement in classification performance following therapy. This suggests some degree of effective differentiation between responders and non-responders, even though it is not seen on the ROC curve.

## 4. Discussion

The results of our study revealed that kurtosis was the only histogram parameter showing significant correlation with treatment response, as post-treatment kurtosis values were significantly higher in responders compared to non-responders (*p* = 0.024), and gained a classification metric, F1 score of 0.821, representing the primary novel finding of this investigation.

While few of other measured histogram parameters (mean, median, 95th percentile) showed statistically significant difference in values between pre- and post-treatment MRI examinations reflecting changes that occur inside tumor tissue as a result of therapy, none of them (mean, standard deviation, skewness, entropy, 5th and 95th percentiles), showed significant differences between responder groups (all *p* > 0.05). This selectivity suggests that kurtosis may be uniquely sensitive to the specific microstructural changes occurring with successful chemoradiotherapy response [15,16].

The biological interpretation of kurtosis relates to tissue heterogeneity, with higher values associated with more concentrated pixel intensity distributions. In the context of treatment response, our finding that responders exhibit higher post-treatment kurtosis values suggests that successful chemoradiotherapy creates more organized tissue patterns as tumor cells are replaced by fibrous tissue, resulting in more concentrated signal intensity distributions [15,16].

Recent studies have extensively investigated histogram analysis in rectal cancer. Azamat et al. [18] explored histogram parameters using T2-weighted sequences and ADC maps, showing that skewness was the parameter with a significant difference between groups with complete and non-complete response. Likewise, our study investigated the use of T2-weighted histogram analysis, but found no significant differences in skewness while demonstrating the significance of kurtosis in predicting response. 

Additionally, Aker et al. [24] in their study conducted on 114 patients performed T2-weighted texture analysis, which captures spatial relationships and patterns within the tissue, finding skewness, mean, and SD capable of detecting CR, whereas our study employed first-order histogram analysis, where parameters were calculated directly from raw signal intensities, treating all voxels independently. Kurtosis, as a measure of distribution peakedness, appears to be the most robust to these methodological differences. Rather than viewing our findings as contradictory to those of Aker et al., we interpret them as complementary. Different analytical approaches—more complex texture analysis versus pure histogram analysis—capture distinct but related aspects of tissue heterogeneity. Future research combining multiple analytical methods may provide superior predictive performance.

It is important to highlight that the use of T2-weighted sequences as an approach enables obtaining valuable information from conventional morphological sequences without requiring more specialized diffusion imaging protocols. This represents a significant practical advantage for clinical implementation [18,19].

De Cecco et al. [25] conducted pioneering work in MRI T2-weighted texture analysis, demonstrating that pre-treatment kurtosis was the best predictor of tumor response with an AUC of 0.907 for discriminating complete pathological response from partial/non-response. However, their study focused on pre-treatment prediction using texture analysis with filtration, while our study demonstrated the value of post-treatment kurtosis assessment using direct T2-weighted histogram analysis. Moreover, Jiménez de Los Santos et al. [26] demonstrated that ADC-derived histogram parameters, especially kurtosis and skewness, represent relevant biomarkers for predicting nCRT response in LARC patients. Their study showed that post-nCRT kurtosis achieved the best diagnostic performance with AUC = 0.985. Similarly, Babaturk et al. [27] found that among ADC histogram parameters of minimum, maximum, 10th, 25th, 50th, 75th, and 90th percentile, mean, and skewness, skewness was the one with the best diagnostic performance, with an achieved AUC of 0.851 for detecting complete response. They analyzed apparent diffusion coefficient histograms, which differ fundamentally from our T2-weighted approach, as ADC and T2-weighted imaging measure distinct tissue properties. ADC reflects water molecule diffusion, with low values reflecting restricted diffusion seen in viable tumor and dense fibrosis, and high ADC values indicating free diffusion associated with necrosis and edema. Conversely, T2-weighted imaging reflects tissue composition and water content, where successful treatment response manifests as homogeneous fibrosis, resulting in more peaked distributions (higher kurtosis) rather than asymmetric tails. We believe that these findings are not contradictory; rather, they demonstrate that different MRI sequences capture different biological processes of treatment response. This observation supports the hypothesis that multiparametric assessment combining T2-weighted kurtosis with ADC-derived parameters would likely achieve superior predictive performance compared to either approach alone.

The value of ADC histogram parameters was also shown by Choi et al. [28], who demonstrated that ADC histogram analysis could differentiate pathological complete response, with 25th percentile ADC showing optimal diagnostic performance. On the contrary, our study did not show any difference between groups in exanimated percentile parameters.

The importance of kurtosis was also stressed by Cui et al. [14], who investigated whole-tumor diffusion kurtosis imaging histogram analysis and found positive correlations between kurtosis parameters and important prognostic factors, including nodal involvement and histological grade. 

Recently, Zhang et al. [16] conducted an important study utilizing advanced diffusion techniques, specifically intravoxel incoherent motion (IVIM) and diffusion kurtosis imaging (DKI). In contrast, our study employed standard T2-weighted sequences that are universally available as part of routine clinical protocols. We view these approaches as serving complementary rather than competitive roles. Zhang et al. demonstrate what is achievable with advanced techniques and specialized expertise, while our study establishes what can be accomplished using widely accessible standard sequences without additional scanning time or cost. This distinction has important implications for health equity and global implementation of quantitative imaging biomarkers. Ultimately, optimal clinical practice may involve a tiered approach, universal baseline assessment using T2-weighted histogram analysis (as demonstrated in our study), with selective application of advanced techniques for complex or borderline cases at institutions with appropriate capabilities.

Our study demonstrated that T2-weighted kurtosis may serve as an independent predictive parameter that might be extracted from standard T2-weighted MRI sequences without requiring additional scanning time. T2-weighted sequences form the foundation of rectal cancer MRI protocols, providing superior tissue contrast for morphological assessment compared to diffusion imaging, which is more susceptible to artifacts that reduce image quality. This approach offers significant advantages over often complex diffusion imaging and multiparametric methods by utilizing readily available, universally accessible imaging data to generate predictive information.

The limitations of this study include a small sample size, imbalanced datasets, a single-center design, and a single radiologist performing manual segmentation, despite acceptable intra-observer reliability, which may introduce individual interpretation bias that could be mitigated by independent evaluation from a second radiologist to assess inter-observer agreement. The study cohort may not be representative of the broader patient population, limiting the generalizability of our findings to different demographic groups or treatment protocols. Additionally, the lack of an external validation cohort prevents assessment of model performance in independent datasets from other institutions with different MRI scanners and acquisition protocols. Future prospective multicenter validation studies should focus on validation of T2-weighted kurtosis thresholds in larger, well-balanced cohorts with external validation datasets to confirm the robustness and generalizability of these findings. Potential integration with AI-based analysis for automated measurement could reduce subjectivity and improve efficiency in clinical workflows [29]. Furthermore, as the results of our previous study have shown [30], the combination of different MRI-gained parameters may increase in MRI-based response prediction. Combining kurtosis with other quantitative biomarkers, including diffusion-weighted imaging (DWI), dynamic contrast-enhanced (DCE) MRI, and radiomics features, should be the subject of future research for the development of comprehensive multiparametric predictive models. Finally, cost-effectiveness analysis and impact on clinical decision-making should be evaluated before widespread adoption of T2-weighted kurtosis imaging in routine clinical practice.

## 5. Conclusions

T2-weighted kurtosis may provide additional benefit in predicting pathological response to neoadjuvant chemoradiotherapy in locally advanced rectal cancer. The integration of T2-weighted kurtosis into routine MRI assessment protocols could enhance precision in rectal cancer treatment response prediction, providing clinicians with an objective, quantitative biomarker that complements existing morphological assessment methods. Validation of these findings through future multicenter studies is necessary to enable their implementation into routine clinical practice.

## Figures and Tables

**Figure 1 biomedicines-13-03003-f001:**
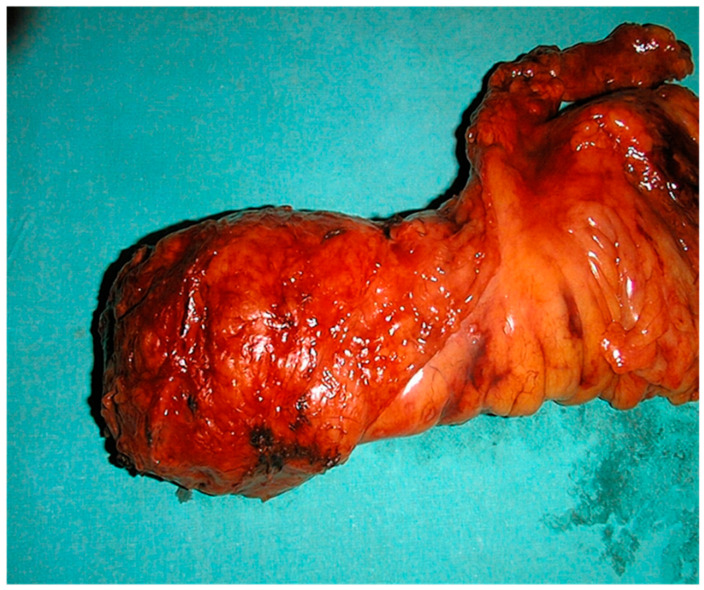
Surgical specimen of total mesorectal excision (TME).

**Figure 2 biomedicines-13-03003-f002:**
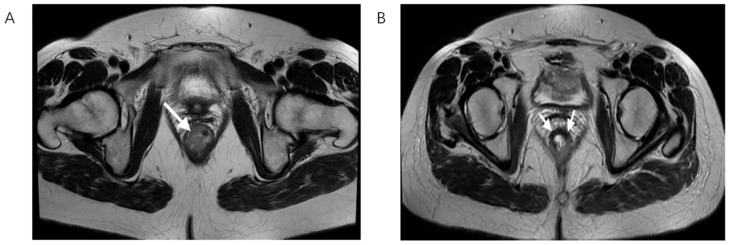
T2-weighted MRI scan in the axial plane showing rectal cancer before (**A**) and after nCRT (**B**), white arrows showing tumor bed. On the left, there is the lesion with its typical intermediate signal intensity, while on the right, a drop in the signal intensity is shown, caused by therapy-induced fibrosis.

**Figure 3 biomedicines-13-03003-f003:**
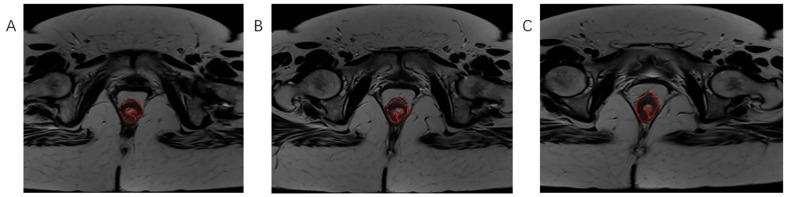
The process of tumor tissue segmentation by manually outlining the ROI on successive slices of T2-weighted axial sequences on MRI examination (**A**–**C**).

**Figure 4 biomedicines-13-03003-f004:**
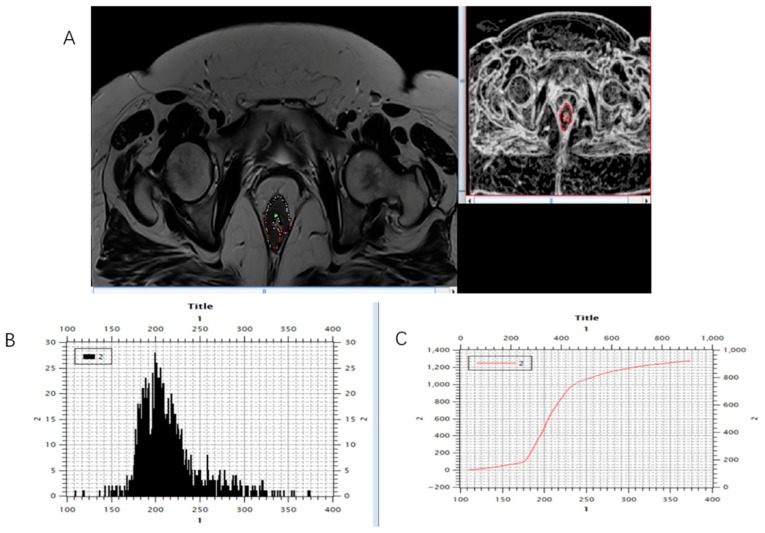
Marking (**A**) and calculating (**B**,**C**) histogram parameters were shown on the initial, pre-treatment MRI scan, T2-weighted axial sequence.

**Figure 5 biomedicines-13-03003-f005:**
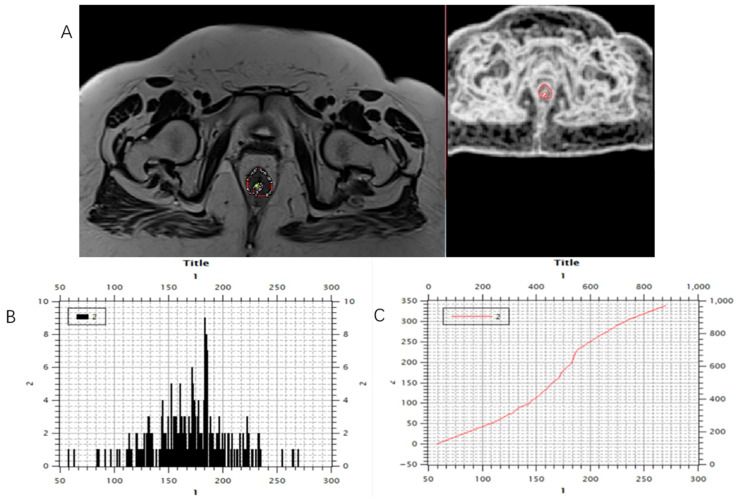
Marking (**A**) and calculating (**B**,**C**) histogram parameters showed on the post-treatment MRI scan, T2-weighted axial sequence.

**Figure 6 biomedicines-13-03003-f006:**
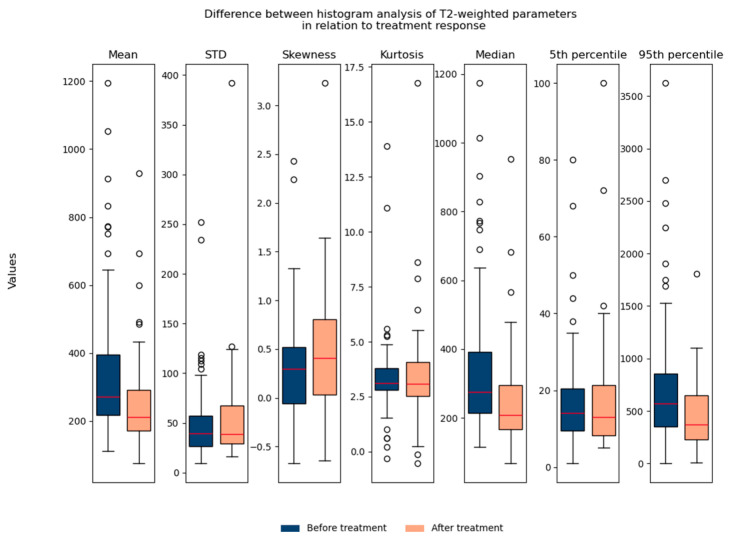
Shows the difference between T2-weighted histogram parameters in relation to treatment response (before and after therapy). STD: standard deviation.

**Figure 7 biomedicines-13-03003-f007:**
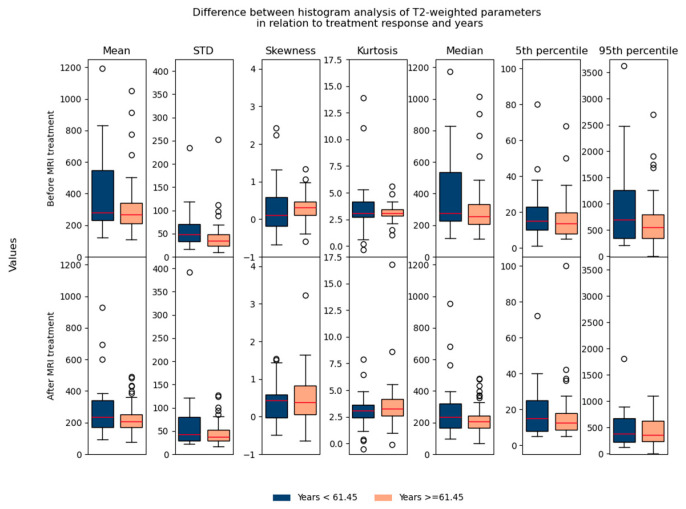
Shows the difference between T2-weighted histogram parameters in relation to treatment response and in relation to age (before and after 61.45 years).

**Figure 8 biomedicines-13-03003-f008:**
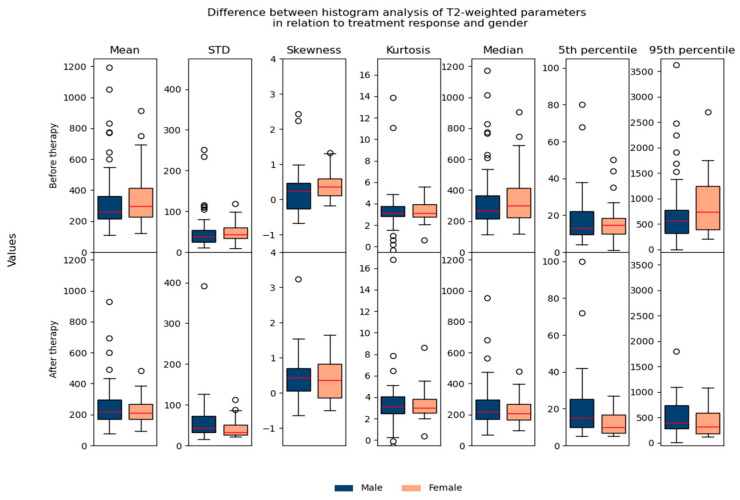
Shows the difference between T2-weighted histogram parameters in relation to treatment response and gender.

**Figure 9 biomedicines-13-03003-f009:**
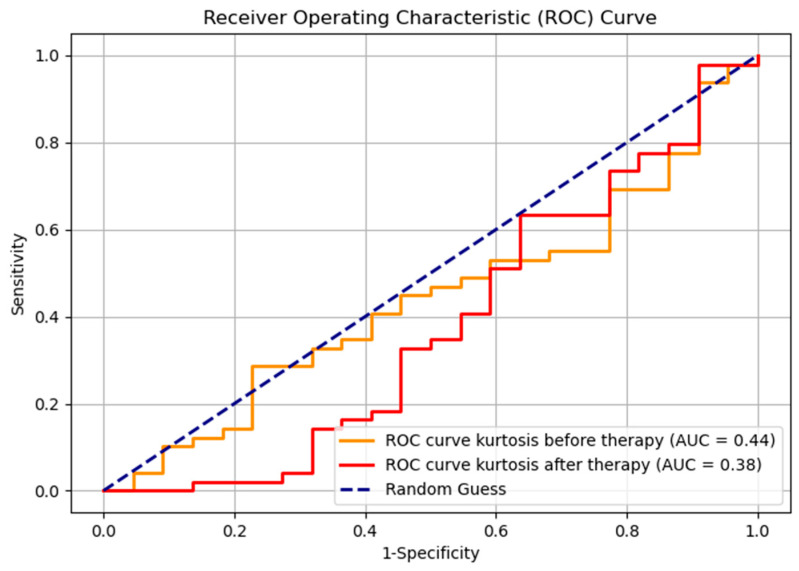
ROC curve for kurtosis before and after therapy, for two groups, responders and non-responders.

**Table 1 biomedicines-13-03003-t001:** T2-weighted histogram parameters in relation to treatment response.

mrTRG	Outcome		*p*-Value
	Non-Responders (*n* = 49) Mean (SE)	Responders (*n* = 22) Mean (SE)	
Before nCRT			
Arithmetic mean	366.88 (34.91)	328.85 (37.54)	0.514
Standard deviation	53.90 (6.70)	43.56 (5.96)	0.341
Skewness	0.26 (0.08)	0.36 (0.14)	0.524
Kurtosis	3.28 (0.22)	3.67 (0.54)	0.430
Median	363.99 (34.28)	325.34 (37.10)	0.500
5th percentile	18.13 (2.11)	16.73 (2.34)	0.691
95th percentile	827.83 (102.58)	658.0 (101.89)	0.315
After nCRT			
Arithmetic mean	255.40 (21.02)	249.88 (28.35)	0.881
Standard deviation	55.89 (7.96)	51.84 (6.98)	0.752
Skewness	0.37 (0.07)	0.62 (0.19)	0.120
Kurtosis	3.01 (0.17)	4.28 (0.73)	**0.024**
Median	250.65 (20.89)	245.34 (27.71)	0.884
5th percentile	16.42 (1.66)	19.95 (4.44)	0.363
95th percentile	475.60 (46.02)	425.29 (58.04)	0.526

TRG: tumor regression grade; nCRT: neoadjuvant chemoradiotherapy; SE: standard error.

**Table 2 biomedicines-13-03003-t002:** Evaluation metrics for kurtosis before and after therapy.

	AUC	Sensitivity	Specificity	PPV	NPV	F1	Acc
Before therapy	0.442	0.286	0.773	0.737	0.327	0.412	0.437
After therapy	0.379	0.980	0.091	0.706	0.667	0.821	0.704

AUC, area under curve; PPV, positive predictive value; NPV, negativne predictive value; Acc, accuracy.

## Data Availability

The raw data supporting the conclusions of this article will be made available by the authors on request.

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
