# Peer review of "Potential Role of T2-Weighted Kurtosis in Improving Response Prediction of Locally Advanced Rectal Cancer as Additional Tool Gained from Standard MRI Examination"

_biomedicines, 2025, doi:10.3390/biomedicines13123003_

Round 1
Reviewer 1 Report
Comments and Suggestions for Authors
Potential role of T2-weighted kurtosis in improving response prediction of locally advanced rectal cancer as additional tool gained from standard MRI examination
The manuscript is interesting, and you can find my suggestions and concerns, section by section as follows:
Introduction: I suggest adding more information on the sex and age of the patients. Similarly, I suggest adding more information about the epidemiology of the LARC. Although the section is quite informative, I suggest adding more about the clinical background, in particular about managing with MRI of LARC; more information on the state of the art is needed. Please, add more information to the introduction. A definition of the analysis that involves histogram metrics is also needed (Please check: Nougaret, S., Vargas, H. A., Lakhman, Y., Sudre, R., Do, R. K., Bibeau, F., ... & Guiu, B. (2016). Intravoxel incoherent motion–derived histogram metrics for assessment of response after combined chemotherapy and radiation therapy in rectal cancer: initial experience and comparison between single-section and volumetric analyses. Radiology, 280(2), 446-454. Not mine and only a suggestion.). The description of the Kurtosis is fine, but I suggest being more statistical and adding its significance for clinical imaging wth MRI.
Methods: The section is clear. I suggest MRI data acquisition instead of MRI technique. Please, add the MRI acquisition parameters used. This is important for replication, despite the existence of protocols.
Figure 3 needs to be improved in quality. Please clarify if the segmentation was performed by expert radiologists. The statistical analysis is well described. Add more information about the specific Python library used. I also suggest adding a brief conceptual explanation of the reason for the statistical analysis chosen.
Results: The results are well-reported. However, my principal concern is about the AUC, which did not reach optimal values. Despite the F1 value after therapy is good, there is a discrepancy with the “before” value.
Discussion: In the discussion, you were able to integrate your results with those previously published. Indeed, as you stated, your study has several limitations that were acknowledged, suggesting a possible explanation and future directions.
Author Response
Manuscript Title: Potential role of T2-weighted kurtosis in improving response prediction of locally advanced rectal cancer as additional tool gained from standard MRI examination
Dear Reviewer,
We sincerely thank the reviewers for their thoughtful and constructive comments on our manuscript. We have carefully considered each suggestion and have made substantial revisions to address all concerns. Below, we provide detailed point-by-point responses to each comment, with changes highlighted in the revised manuscript.
Reviewer 1.
Comment 1: I suggest adding more information on the sex and age of the patients. Similarly, I suggest adding more information about the epidemiology of the LARC.
Response: We thank the reviewer for this suggestion. We have expanded the introduction to include more comprehensive epidemiological information about LARC, including sex distribution and age demographics. Specifically, we have added:
Rectal cancer shows a male predominance with a male-to-female ratio of approximately 1.5:1, with median age at diagnosis ranging from 60-65 years. The incidence of LARC accounts for approximately 30-40% of all rectal cancer cases.
Changes made: Introduction, lines 44-47
Comment 2: Although the section is quite informative, I suggest adding more about the clinical background, in particular about managing with MRI of LARC; more information on the state of the art is needed.
Response: We agree that the clinical context deserves more elaboration. We have added to the discussion of MRI's role in LARC management:
Magnetic resonance imaging (MRI) is pivotal in managing LARC with its roles for primary staging, restaging, and surveillance. It has extraordinary accuracy in the initial, pre-treatment examination, offering comprehensive insights for local staging. The European Society for Medical Oncology (ESMO) guidelines recommend high-resolution MRI as the imaging modality of choice for rectal cancer staging. Unfortunately, it is worth highlighting that, in contrast to the great precision of the initial above-mentioned magnetic resonance scan, the reliability of interpretation of the post-treatment MRI drops significantly, with reported poor agreement in staging between post-nCRT MRI and histopathological finding. The reason of this drop is primarily attributed to the difficulty in differentiating therapy-induced fibrosis, edema and inflammation from residual tumor tissue. This limitation has important clinical implications, particularly for patient selection for organ preservation strategies such as ‘watch and wait’ approach, where accurate identification of complete responders is essential.
Changes made: Introduction, lines 56-68. Referances added.
Comment 3: A definition of the analysis that involves histogram metrics is also needed. Please check: Nougaret, S., et al. (2016). Intravoxel incoherent motion–derived histogram metrics for assessment of response after combined chemotherapy and radiation therapy in rectal cancer: initial experience and comparison between single-section and volumetric analyses. Radiology, 280(2), 446-454.
Response3: We appreciate this valuable suggestion and the recommended reference. We have added a comprehensive definition of histogram analysis in medical imaging and its clinical relevance:
Histogram analysis represents a quantitative approach to medical image interpretation that extracts statistical information from pixel intensity distributions within regions of interest. It examines how frequently different signal intensity values appear across pixels or voxels within a specified region or volume of interest. Rather than relying solely on average intensity measurements, this technique analyzes the entire range of intensity values and their distribution characteristics. It generates various statistical metrics—including mean, median, standard deviation, skewness, kurtosis, and percentile values—that characterize tissue variability.This methodology can be utilized across different MRI sequences and has demonstrated potential in cancer imaging for evaluating tumor features and monitoring therapeutic response. The key benefit of this approach is its capacity to measure tumor heterogeneity in an objective and consistent manner, potentially detecting subtle variations that might escape detection through visual interpretation or standard quantitative analysis.
Changes made: Introduction. Lines 83-95.
Comment 4: The description of the Kurtosis is fine, but I suggest being more statistical and adding its significance for clinical imaging with MRI.
Response4: We have enhanced the description of kurtosis with more statistical detail and clinical context:
Kurtosis is a statistical measure that quantifies the “peakedness” or “tailedness” of a probability distribution, providing information about the concentration of values around the distribution center. In the context of medical imaging, kurtosis reflects tissue heterogeneity, with higher values indicating more concentrated pixel intensity distributions and lower values suggesting more dispersed, heterogeneous patterns. In other words, kurtosis reflects the degree of tissue heterogeneity at the microstructural level. Higher kurtosis values indicate more homogeneous tissue with pixel intensities tightly clustered around the mean, which in treated rectal cancer may correspond to uniform fibrotic tissue replacement following successful therapy. Conversely, lower kurtosis values suggest greater heterogeneity, potentially indicating admixed viable tumor, necrosis, and treatment effects. Recent studies have demonstrated that kurtosis and other histogram parameters derived from diffusion-weighted and dynamic contrast-enhanced imaging show promise in predicting treatment response in rectal cancer.
Changes made: Introduction, lines 99-109.
Comment 5: I suggest MRI data acquisition instead of MRI technique. Please, add the MRI acquisition parameters used. This is important for replication.
Response5: We agree this terminology is more appropriate and that complete acquisition parameters are essential for reproducibility. We have retitled the section to "2.2. MRI Acquisition" and added comprehensive technical parameters:
Complete acquisition parameters now included:
All patients were examined using a 1.5 Tesla MR scanner (Signa HDxt; GE Healthcare, Milwaukee, Wisconsin, USA) before and after neoadjuvant chemoradiotherapy. Patient preparation included enema for rectal lumen cleansing and spasmolytic agent hyoscine-butyl bromide (Buscopan) 20 mg intravenously. They were placed in a supine position, and a phased-array body coil was used for imaging. The MRI protocol consisted of axial and sagittal fast spin echo (FSE) T2-weighted imaging (TR/TE = 4500/100 ms; FOV: 280 × 280 mm; matrix: 384 × 256; slice thickness: 4 mm; spacing: 1 mm); oblique para-axial (perpendicular to the longitudinal axis of the tumor) FSE T2-weighted imaging (TR/TE = 5000/110 ms; matrix = 512 × 512; FOV = 180 × 180 mm; slice thickness = 3 mm; spacing: 0 mm); para-coronal FSE T2-weighted imaging (TR/TE = 5000/110 ms; matrix = 384 × 384; FOV = 200 × 200 mm; slice thickness = 3 mm; spacing: 0 mm); and para-axial diffusion-weighted imaging (TR/TE = 7000/70 ms; matrix = 128 × 128; FOV = 280 × 280 mm; slice thickness: 5 mm; spacing: 1 mm; b-values = 0, 2000 s/mm²; NEX = 6).​​​​​​​​​​​​​​​​
Changes made: Methods section 2.2, MRI acquisition, lines 136-151
Comment 6: Figure 3 needs to be improved in quality. Please clarify if the segmentation was performed by expert radiologists.
Response6: We apologize for the suboptimal image quality. We have replaced Figure 3 with higher resolution images. Regarding segmentation, we have added the following clarification:
All tumor segmentations were performed by board-certified radiologist with 8 years of experience in rectal cancer imaging, blinded to pathological results. Segmentation was performed on T2-weighted axial sequences with regions of interest manually drawn on each slice containing visible tumor while carefully excluding surrounding fat, mesorectal fascia, and bowel lumen. It was followed by volume of interest (VOI) creation and automatically histogram parameter calculation (Figure 3-5). The following 7 histogram parameters were measured: mean (average pixel intensity value in the region), standard deviation (measure of pixel intensity variation), median (middle pixel intensity value), kurtosis (measure of distribution “peakedness”), skewness (measure of pixel distribution asymmetry), 5th and 95th percentiles (markers of distribution extremes). Beside above-mentioned, other measured morphological, functional and volumetric parameters are beyond the scope of this study.
To ensure consistency, a standardized segmentation protocol established prior to the study was followed. The radiologist had access to all available imaging sequences and clinical information necessary for accurate delineation. Quality control was performed by the same radiologist. A subset of 30 randomly selected cases was re-segmented after a washout period of 8 weeks to assess intra-observer reliability. The radiologist was blinded to their initial segmentation. Intra-observer agreement was evaluated using inraclass correlation coefficient (ICC) and Dice similarity coefficient (DSC) demonstrating excellent reproducibility (ICC=0.85; DSC= 0.80).
Changes made: Methods section 2.3, MRI technique, lines 163-182
Comment 7:The statistical analysis is well described. Add more information about the specific Python library used. I also suggest adding a brief conceptual explanation of the reason for the statistical analysis chosen.
Response7: Thank you very much for your suggestion. The Section 2.4 Statistical Analysis is edited with detailed information about statistical software and justification as follows:
Descriptive and analytical statistical methods were used in this study. Among the descriptive methods, measures of central tendency (arithmetic mean) and measures of variability (standard deviation, standard error) were applied. Categorical data were presented using absolute and relative frequencies. Independent sample t-tests, as exploratory method, were employed to compare parameters between responder and non-responder groups, appropriate for comparing two independent groups with continuous variables. Diagnostic accuracy involved sensitivity, specificity, positive and negative predictive values and F1 score. In all analyses, the level of statistical significance was set at p ≤ 0.05. Statistical analyses were performed using Python 3.10. with the usage of following libraries: SciPy for statistical tests, scikit-learn and stastmodels.api for metrics, as well as NumPy, pandas and matplotlib for data management and visualization.
Changes made: Methods section 2.4 Statistical analysis lines 238-247
Comment 8:Results: The results are well-reported. However, my principal concern is about the AUC, which did not reach optimal values. Despite the F1 value after therapy is good, there is a discrepancy with the “before” value.
Response 8: We totally agree that there is concern about the AUC, which did not reach optimal values, but as we already stated, and now edited more briefly Section 3.2:
The Receiver Operating Characteristic (ROC) curve, with Youden index used for selecting optimal cutoff point, for kurtosis before and after therapy demonstrates the results that are lower compared to random classification (Figure 9). However, such outcomes are not uncommon in the context of imbalanced datasets (responders – 22, non-responders – 49, ration 1:2.2). This phenomenon can occur because the ROC metric may be less robust in scenarios where class distributions are highly skewed, it gives equal weight to both classes regardless of their prevalence, therefor it is suboptimal for visual representation. The evaluation metrics for the classification of responders and non-responders, both before and after therapy was showed in Table 2. Notably, the F1 score as a metric that accounts imbalanced data, indicates an improvement in classification performance following therapy. This suggests some degree of effective differentiation between responders and non-responders, even though it is not seen on ROC curve.
Changes made: Results 3.2 lines 280, 283, 285
Comment 9:In the discussion, you were able to integrate your results with those previously published. Indeed, as you stated, your study has several limitations that were acknowledged, suggesting a possible explanation and future directions.
Response9: We thank the reviewer for this positive feedback. We have further strengthened the limitations section and expanded future directions to include:
The limitations of this study include a small sample size, imbalanced datasets, single-center design and a single radiologist performing manual segmentation, despite acceptable intra-observer reliability, may introduce individual interpretation bias that could be mitigated by independent evaluation from a second radiologist to assess inter-observer agreement. The study cohort may not be representative of the broader patient population, limiting the generalizability of our findings to different demographic groups or treatment protocols. Additionally, the lack of external validation cohort prevents assessment of model performance in independent datasets from other institutions with different MRI scanners and acquisition protocols. Future prospective multicenter validation studies should focus on validation of T2-weighted kurtosis thresholds in larger, well-balanced cohorts with external validation datasets to confirm the robustness and generalizability of these findings. Potential integration with AI-based analysis for automated measurement could reduce subjectivity and improve efficiency in clinical workflows [32]. Furthermore, as the results of our previous study have shown [33], the combination of different MRI gained parameters may increase the MRI-based response prediction. Combining kurtosis with other quantitative biomarkers, including diffusion-weighted imaging (DWI), dynamic contrast-enhanced (DCE) MRI, and radiomics features, should be the subject of future research for development of comprehensive multiparametric predictive models. Finally, cost-effectiveness analysis and impact on clinical decision-making should be evaluated before widespread adoption of T2-weighted kurtosis imaging in routine clinical practice.​​​​​​​​​​​​​​​​
Changes made: 4. Discussion, lines 361-391
Sincerely,
Aleksandra Jankovic et al.

Reviewer 2 Report
Comments and Suggestions for Authors
The authors retrospectively assessed T2-weighted histogram parameters including kurtosis in 71 LARC patients who underwent nCRT and surgery. This study reported that post-treatment T2 kurtosis was significantly higher in responders than non-responders, proposing T2 kurtosis as a potential quantitative biomarker extractable from standard T2 MRI without extra scanning time. This clinically important topic could improve noninvasive prediction of response after nCRT in rectal cancer that the utility of T2-weighted histogram kurtosis as a practical biomarker seemed interesting concerning its widely available sequences. However, the manuscript has several limitations that must be addressed before publication as following.
- The cohort wassmall with strong imbalance (responders n=22 non-responders n=49). Although this imbalance undermining the stability of diagnostic performance metrics has been acknowledged, I think the author should perform additional analyses accounting for imbalance, including precision-recall curves and average precision, and bootstrapped confidence intervals for AUC to estimate robustness.
- The manuscript tested7 histogram parameters at pre and post timepoints and stratified by outcome. No multiple comparison correction has been The finding of p = 0.024 for post-treatment kurtosis might not survive correction. Please use Benjamini-Hochberg or Bonferroni method as an appropriate correction for multiple testing and report adjusted p-values.
- The AUCs were reportedlow (0.379-442), but F1 and sensitivity after therapy were reported high (sensitivity: 0.98 and F1: 0.821). This inconsistency likely attributed to threshold selection and class imbalance, so AUC might be misleading. How thresholds were chosen for sensitivity/specificity? Was Youden index used, or a fixed threshold? Please explain the apparent contradiction clearly.
- Tumor segmentation was performed manually using MIPAV by outlining ROIs on axial T2 slices. Since manual segmentation is operator-dependent, the authors should report: number of readers,reader expertise, whether segmentation was performed blinded to outcome, and inter-observer and intra-observer Dice and ICC for histogram parameters.
- Histogram measures weresensitive to acquisition settings, intensity normalization, and preprocessing, I suggest the authors to provide full T2 acquisition parameters and describe image preprocessing steps, including noise reduction and bias field correction. Did histogram perform normalization? If no, please discuss more about how interscan/amplitude variations might affect kurtosis.
- Several confounding factors that might influencekurtosis have not been controlled, such as tumor size/volume, time interval between end of nCRT and MRI, use of antispasmodics/enema, etc. It’s better to perform multivariable logistic regression including important potentially confounding
Author Response
Manuscript Title: Potential role of T2-weighted kurtosis in improving response prediction of locally advanced rectal cancer as additional tool gained from standard MRI examination
Dear Reviewer,
We sincerely thank the reviewers for their thoughtful and constructive comments on our manuscript. We have carefully considered each suggestion and have made substantial revisions to address all concerns. Below, we provide detailed point-by-point responses to each comment, with changes highlighted in the revised manuscript.
Reviewer 2.
Comment 1:The cohort was small with strong imbalance (responders n=22 non-responders n=49). Although this imbalance undermining the stability of diagnostic performance metrics has been acknowledged, I think the author should perform additional analyses accounting for imbalance, including precision-recall curves and average precision, and bootstrapped confidence intervals for AUC to estimate robustness.
Response1: Thank you for the comment. As our analysis was exploratory and the primary aim was to describe the behavior of individual variables in responders and non-responders, we performed ROC-AUC curve and classification metrics, we did not perform precision–recall analysis, or bootstrapped confidence intervals. Given the limited sample size and class imbalance, these metrics would provide unstable or uninterpretable estimates. Therefore, we rely only on descriptive classification metrics (e.g., F1-score and sensitivity), without making inferential claims about model performance.
Comment 2:The manuscript tested 7 histogram parameters at pre and post timepoints and stratified by outcome. No multiple comparison correction has been made. The finding of p = 0.024 for post-treatment kurtosis might not survive correction. Please use Benjamini-Hochberg or Bonferroni method as an appropriate correction for multiple testing and report adjusted p-values.
Thank you very much for your valuable comment. Although several variables were initially tested, the analyses were exploratory. Also, the tested variables were analyzed separately, we looked between two groups (responders and non-responders) s the multiple testing was not conducted. Therefore, we did not apply multiple testing correction (e.g., Benjamini–Hochberg or Bonferroni).
The purpose of the univariate step was to identify potential candidate variables. Variables that reached p<0.05 were subsequently evaluated in the predictive model, where their performance was assessed using the F1-score, which represents an independent validation metric and does not rely on inferential statistical significance.
For this reason, we interpret p-values descriptively and avoid drawing strict inferential conclusions from them, we edited whole manuscript to be clearer about the p_value.
Comment3: The AUCs were reported low (0.379-442), but F1 and sensitivity after therapy were reported high (sensitivity: 0.98 and F1: 0.821). This inconsistency likely attributed to threshold selection and class imbalance, so AUC might be misleading. How thresholds were chosen for sensitivity/specificity? Was Youden index used, or a fixed threshold? Please explain the apparent contradiction clearly.
Thank you for your insights, and yes, the inconsistency is likely attributed to class imbalance. The threshold that is used for AUC was Youden index, as it provides a data-driven approach to selecting an optimal cutoff point. We added that in Section 3.2.:
The Receiver Operating Characteristic (ROC) curve, with Youden index used for selecting optimal cutoff point, for kurtosis before and after therapy demonstrates the results that are lower compared to random classification (Figure 9). However, such outcomes are not uncommon in the context of imbalanced datasets (responders – 22, non-responders – 49, ration 1:2.2). This phenomenon can occur because the ROC metric may be less robust in scenarios where class distributions are highly skewed, it gives equal weight to both classes regardless of their prevalence, therefor it is suboptimal for visual representation. The evaluation metrics for the classification of responders and non-responders, both before and after therapy was showed in Table 2. Notably, the F1 score as a metric that accounts imbalanced data, indicates an improvement in classification performance following therapy. This suggests some degree of effective differentiation between responders and non-responders, even though it is not seen on ROC curve.
Changes made: Results 3.2 lines 280, 283, 285
Coment 4:Tumor segmentation was performed manually using MIPAV by outlining ROIs on axial T2 slices. Since manual segmentation is operator-dependent, the authors should report: number of readers, reader expertise, whether segmentation was performed blinded to outcome, and inter-observer and intra-observer Dice and ICC for histogram parameters.
Response4: This is an essential methodological detail that we should have reported more thoroughly. We have now added comprehensive information about the segmentation protocol and reproducibility analysis:
All tumor segmentations were performed by board-certified radiologist with 8 years of experience in rectal cancer imaging, blinded to pathological results. Segmentation was performed on T2-weighted axial sequences with regions of interest manually drawn on each slice containing visible tumor while carefully excluding surrounding fat, mesorectal fascia, and bowel lumen. It was followed by volume of interest (VOI) creation and automatically histogram parameter calculation (Figure 3-5). The following 7 histogram parameters were measured: mean (average pixel intensity value in the region), standard deviation (measure of pixel intensity variation), median (middle pixel intensity value), kurtosis (measure of distribution “peakedness”), skewness (measure of pixel distribution asymmetry), 5th and 95th percentiles (markers of distribution extremes). Beside above-mentioned, other measured morphological, functional and volumetric parameters are beyond the scope of this study.
To ensure consistency, a standardized segmentation protocol established prior to the study was followed. The radiologist had access to all available imaging sequences and clinical information necessary for accurate delineation. Quality control was performed by the same radiologist. A subset of 30 randomly selected cases was re-segmented after a washout period of 8 weeks to assess intra-observer reliability. The radiologist was blinded to their initial segmentation. Intra-observer agreement was evaluated using inraclass correlation coefficient (ICC) and Dice similarity coefficient (DSC) demonstrating excellent reproducibility (ICC=0.85; DSC= 0.80).
Changes made: Methods section 2.3, MRI technique, lines 163-182
Comment 5: Histogram measures were sensitive to acquisition settings, intensity normalization, and preprocessing. I suggest the authors to provide full T2 acquisition parameters and describe image preprocessing steps, including noise reduction and bias field correction. Did histogram perform normalization? If no, please discuss more about how interscan/amplitude variations might affect kurtosis.
Response5: This is a crucial technical concern that affects the validity and reproducibility of our findings. We have now provided comprehensive details:
The MRI protocol consisted of axial and sagittal fast spin echo (FSE) T2-weighted imaging (TR/TE = 4500/100 ms; FOV: 280 × 280 mm; matrix: 384 × 256; slice thickness: 4 mm; spacing: 1 mm); oblique para-axial (perpendicular to the longitudinal axis of the tumor) FSE T2-weighted imaging (TR/TE = 5000/110 ms; matrix = 512 × 512; FOV = 180 × 180 mm; slice thickness = 3 mm; spacing: 0 mm); para-coronal FSE T2-weighted imaging (TR/TE = 5000/110 ms; matrix = 384 × 384; FOV = 200 × 200 mm; slice thickness = 3 mm; spacing: 0 mm); and para-axial diffusion-weighted imaging (TR/TE = 7000/70 ms; matrix = 128 × 128; FOV = 280 × 280 mm; slice thickness: 5 mm; spacing: 1 mm; b-values = 0, 2000 s/mm²; NEX = 6).
The normalization of histogram does not affect the histogram shape or its peakedness and thus kurtosis and skewness remain the same whether the procedure was performed or not. The acquisition parameters of T2 sequence were the same for all subjects and they are now introduced in Materials and methods selection. Regarding the reprocessing procedures, we chose not to apply them because they could add confounds to analysis.
Changes made: Methods section 2.2, MRI acquisition, lines 136-151
Comment 6: Several confounding factors that might influence kurtosis have not been controlled, such as tumor size/volume, time interval between end of nCRT and MRI, use of antispasmodics/enema, etc. It's better to perform multivariable logistic regression including important potentially confounding factors.
Response6: We acknowledge that several clinical and technical factors could theoretically affect kurtosis values, including tumor size/volume, time interval between completion of neoCRT and post-treatment MRI, and patient preparation factors (use of antispasmodics, bowel preparation).
In our preliminary analysis, we evaluated these potential confounding variables for their association with kurtosis measurements. However, we did not observe statistically significant relationships between these factors and kurtosis values in our cohort.
Given the absence of significant associations between these variables and kurtosis measurements in our dataset, multivariable logistic regression analysis was not performed, as the primary analysis focused on the direct relationship between kurtosis and treatment response.
We recognize that our single-center design with standardized acquisition protocols and patient preparation may have inherently minimized variability from these confounding factors. However, we agree that future multicenter studies with larger, more heterogeneous populations should systematically evaluate these variables and consider multivariable modeling approaches to confirm the independent predictive value of kurtosis parameter.
We have addressed this limitation in the Discussion section and emphasized the need for comprehensive multivariable analysis in future validation studies.​​​​​​​
Changes made: 4. Discussion, lines 361-391
Sincerely,
Aleksandra Jankovic et al.

Round 2
Reviewer 1 Report
Comments and Suggestions for Authors
The authors have addressed the concerns that I have raised. They also replied to my questions satisfactorily.